# WorldPack: Dynamic Frame Compression for Long-context Video World Modeling

## Abstract

Video world models have attracted significant attention for their ability to produce high-fidelity future visual observations conditioned on past observations and navigation actions. However, achieving temporally and spatially consistent generation over long horizons remains an open challenge: existing approaches either compress past frames using generic importance schedules that do not explicitly exploit 3D viewpoint geometry, or retrieve only a handful of spatially relevant frames without increasing the total amount of retained history. In this paper, we propose *WorldPack*, a video world model that introduces spatially-aware compressed memory to address both limitations simultaneously. The key insight is that compression rates should not be uniform or temporally determined, but should instead be dynamically allocated based on 3D spatial relevance to the current viewpoint. WorldPack achieves this through two tightly coupled mechanisms: *trajectory packing*, which fits substantially more historical frames into a fixed-length context through hierarchical frame compression, and *geometric selection*, which leverages camera pose information and field-of-view overlap to assign lower compression to spatially important frames and higher compression to less relevant ones. Together, these mechanisms expand the effective context from 4 to 22 frames with moderate computational overhead: trajectory packing increases diffusion-model inference time by 16%, while FoV-based geometric selection introduces an additional candidate-dependent cost. We evaluate WorldPack on LoopNav, a Minecraft benchmark for long-horizon spatial consistency, and conduct comprehensive experiments on the RECON, real-world navigation dataset, across multiple evaluation protocols. WorldPack outperforms strong baselines—including Oasis, Mineworld, DIAMOND, NWM—with particularly pronounced gains in spatial reasoning tasks that require recall of distant observations.

## 1 Introduction

Video world models, i.e., neural world simulators based on video generation models, have recently attracted significant attention for their ability to produce high-fidelity future visual observations conditioned on past observations and navigation actions (Brooks et al., 2024; Ball et al., 2025; World Labs, 2025a; Hafner et al., 2025). By predicting and generating future visual observations from past observations and agent actions, these models hold the potential to serve as alternatives to conventional simulation environments. Their applications span a wide range of domains, such as robotic simulation (Bar et al., 2024; Hu et al., 2025; Zhu et al., 2025; Mao et al., 2025a; Chen et al., 2025), autonomous driving (Hu et al., 2023; Russell et al., 2025; Wang et al., 2023; Zhao et al., 2024; Gao et al., 2024b), and AI-driven content generation in game engines (Alonso et al., 2024; Valevski et al., 2024; Bruce et al., 2024).

Despite this promise, long-context video world modeling remains challenging (Decart et al., 2024; Guo et al., 2025). A model must retain temporally distant observations over long rollouts while preserving spatially critical details needed to reconstruct previously visited places. However, naively extending the context is computationally expensive (Vaswani et al., 2017; Peebles & Xie, 2023; Oshima et al., 2024; Gao et al., 2024c), whereas fixed or recency-based compression can discard information that is spatially relevant but temporally distant (Decart et al., 2024; Guo et al., 2025).

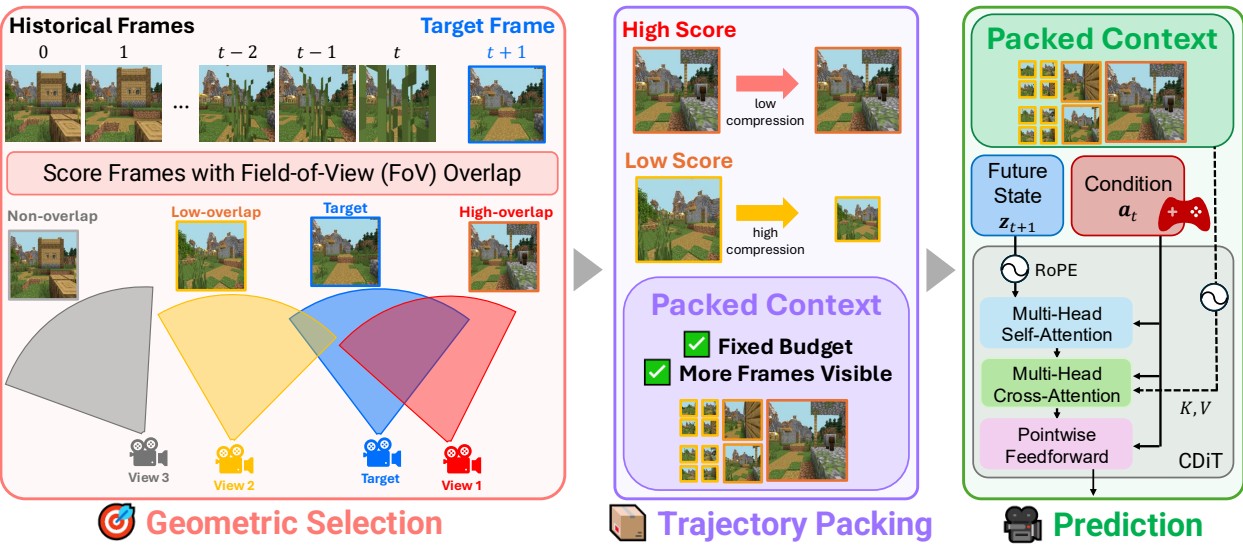

Figure 1: WorldPack consists of (1) geometric selection: dynamic allocation of compression rate based on camera pose information, (2) trajectory packing: packing the trajectory into the context, and (3) CDiT with RoPE-based timestep embedding.

Recent work has begun to address this long-context memory problem from two complementary but disconnected angles. On the one hand, dynamic compression methods allocate varying token budgets or compression rates across frames or image regions to make more efficient use of a fixed visual token budget (Yan et al., 2025; Shen et al., 2026; Zhang & Agrawala, 2025). FramePack (Zhang & Agrawala, 2025) compresses past video frames at varying rates based on frame-wise importance and considers temporal proximity, feature similarity, and hybrid importance measures, none of which explicitly model camera-pose geometry or 3D covisibility. On the other hand, spatial memory retrieval methods (Xiao et al., 2025; Yu et al., 2025a; Wu et al., 2025a) select past frames based on field-of-view overlap or 3D co-visibility, but they operate within a fixed context window, replacing less relevant frames entirely rather than retaining them at reduced resolution.

In this paper, we propose WorldPack, a video world model that bridges these two directions by introducing spatially-aware compressed memory. Rather than treating frame selection and frame compression as independent problems, WorldPack unifies them: it packs many historical frames into a fixed-length context while dynamically allocating compression rates based on 3D spatial relevance. Frames that strongly overlap with the current observation are preserved at high resolution, while less relevant frames are aggressively compressed but still retained, ensuring that no historical information is entirely discarded. This design enables the model to reason over substantially longer horizons without incurring a proportional increase in computational cost.

We build WorldPack on a conditional diffusion transformer (CDiT) (Bar et al., 2024) backbone with RoPE-based (Su et al., 2023) temporal embeddings, and evaluate it on LoopNav (Lian et al., 2025), a Minecraft benchmark for long-horizon spatial consistency, across both spatial memory retrieval and spatial reasoning tasks. We further conduct comprehensive experiments on the RECON dataset (Shah et al., 2021) under multiple protocols to demonstrate effectiveness on real-world data. Through detailed ablation studies, we reproduce the two most closely related approaches within our own backbone as controlled baselines: (i) the temporal-proximity packing of FramePack (Zhang & Agrawala, 2025), and (ii) the spatial retrieval mechanism of WorldMem (Xiao et al., 2025) and Context-as-Memory (Yu et al., 2025a). Comparing WorldPack against (i) isolates the contribution of geometric selection, and against (ii) isolates the contribution of trajectory packing; together these reveal that the two mechanisms address complementary bottlenecks—expanding the amount of available history and determining how that history is compressed.

# 2 Related Work

## 2.1 Video World Models

Recent advances in video diffusion models have enabled photorealistic, high-resolution video generation, positioning them as "general-purpose world simulators" capable of producing diverse scenes with plausible dynamics from text (Ho et al., 2022b;a; Brooks et al., 2024; Google DeepMind, 2024; Kang et al., 2024; Bansal et al., 2024; Chefer et al., 2025; Wu et al., 2025b; Oshima et al., 2025). Building on this progress, video world models have attracted significant attention for their ability to generate high-fidelity future visual observations conditioned on past scene sequences and navigation actions (Ball et al., 2025; World Labs, 2025a; Mao et al., 2025c;b; Hong et al., 2025; HunyuanWorld, 2025; Hafner et al., 2025). Their applications span a wide range of domains, such as game engines (Valevski et al., 2024; Decart et al., 2024; Guo et al., 2025; Bruce et al., 2024), autonomous driving (Hu et al., 2023; Russell et al., 2025; Wang et al., 2023; Zhao et al., 2024; Gao et al., 2024b; Hu et al., 2024; Guo et al., 2024), and robotics (Bar et al., 2024; Zhu et al., 2025; Hu et al., 2025; Mao et al., 2025a; Chen et al., 2025). These applications underscore the importance of maintaining long-term temporal and spatial consistency, particularly in decision-making tasks such as driving and navigation.

However, achieving such coherence remains an unresolved challenge, even for state-of-the-art models, due to the prohibitively high computational costs required to process a long sequence of observations in the model context (Decart et al., 2024; Guo et al., 2025). Recent studies (Yu et al., 2025a; Xiao et al., 2025; World Labs, 2025b) propose spatial retrieval mechanisms that select past frames based on overlapping fields of view, improving spatial consistency by ensuring that relevant observations are included in the context. However, these methods operate within a fixed context window: they choose which frames to include, but cannot increase the total number of frames accessible to the model. As a result, the trade-off between spatial relevance and historical coverage remains unresolved; a spatially relevant frame from the distant past may be included only at the cost of discarding other potentially useful observations.

## 2.2 Long-Context Video Generation

In video generation, extensive research has focused on extending fixed-length generation horizons to long-term rollouts. Representative directions include temporal super-resolution with coarse-to-fine processing (Ho et al., 2022a; Yin et al., 2023), as well as architectural advances aimed at capturing long-range dependencies (Gu et al., 2021; Gu & Dao, 2023; Oshima et al., 2024; Gao et al., 2024c). While these methods enable the generation of longer videos, they ultimately remain constrained by fixed-length outputs. One of the major research directions toward overcoming this limitation is autoregressive long-term video generation. These approaches generate videos sequentially conditioned on recent frames (He et al., 2022; Henschel et al., 2024; Po et al., 2025b; Jin et al., 2024; Kodaira et al., 2025; Yang et al., 2025; Gao et al., 2025; Qiu et al., 2025), and include inference-time techniques that adapt pretrained models to longer rollouts without retraining (Qiu et al., 2023; Kim et al., 2024), as well as few-step model distillation methods (Yin et al., 2025).

However, autoregressive long-term video generation suffers from error accumulation and memory forgetting as the rollout length increases (Wang et al., 2025). To mitigate error accumulation, various stabilization methods have been explored, including combining next-token prediction with full-sequence diffusion (Chen et al., 2024; Ruhe et al., 2024; Song et al., 2025), and training models to correct drift by directly conditioning on their own generated frames during autoregressive rollouts (Huang et al., 2025; Shin et al., 2025; Cui et al., 2025; Po et al., 2025a; Yu et al., 2025b). Recently, Zhang & Agrawala (2025) proposed compressing past frames at varying rates when injecting them into the context, retaining long histories while reducing the impact of accumulated drift. However, their compression schedule is determined by temporal proximity, frame similarity, and their hybrid, in which 3D spatial relevance is not explicitly considered when selecting the most informative frames. In this work, we transfer such context compression techniques to the setting of video world modeling and, specifically, introduce a geometry-based importance measure derived from camera poses and 3D co-visibility.

## 3    Preliminaries

We begin by extending latent diffusion models (Rombach et al., 2022) to the temporal domain, formulating video diffusion models (He et al., 2022; Ho et al., 2022a). Given a sequence of frames $\mathbf{x}_{0:T} = (\mathbf{x}_0, \mathbf{x}_1, \ldots, \mathbf{x}_T)$, we first encode frames into latent representations $\mathbf{z}_{0:T} = (\mathbf{z}_0, \mathbf{z}_1, \ldots, \mathbf{z}_T)$ using a pretrained VAE (Kingma & Welling, 2013), i.e., $\mathbf{z}_i = \text{Enc}(\mathbf{x}_i)$. In this setting, all latent frames share the same noise level $k$, and the reverse diffusion process restores the clean sequence by iteratively denoising:

$$p_\theta(\mathbf{z}_{0:T}^{k-1} \mid \mathbf{z}_{0:T}^k) = \mathcal{N}\big(\mathbf{z}_{0:T}^{k-1}; \mu_\theta(\mathbf{z}_{0:T}^k, k), \sigma_k^2 I\big), \tag{1}$$

where $\mathbf{z}_{0:T}^k$ denotes the noisy latent sequence at noise level $k$. This full-sequence formulation provides global guidance across frames, but constrains the sequence length to that used during training and lacks flexibility for long-horizon rollouts.

To overcome this limitation, we adopt an autoregressive formulation. Instead of generating the entire sequence jointly, the model conditions on the most recent $m$ latent frames to predict the next one:

$$p_\theta(\mathbf{z}_{t+1} \mid \mathbf{z}_{t-m+1:t}), \tag{2}$$

where generation proceeds sequentially. This setup naturally extends video length beyond the training horizon and supports long-term coherent generation.

Finally, to obtain an interactive video world model, we further introduce action sequences into the formulation. Given past latent states $\mathbf{z}_{t-m:t}$ and the current action $\mathbf{a}_t$, we learn a stochastic transition model $F_\theta$:

$$\mathbf{z}_{t+1} \sim F_\theta(\mathbf{z}_{t+1} \mid \mathbf{z}_{t-m:t}, \mathbf{a}_t). \tag{3}$$

This formulation approximates the environment dynamics $p(\mathbf{z}_{t+1} \mid \mathbf{z}_{\leq t}, \mathbf{a}_{\leq t})$, while operating in the compressed latent space. The predicted next state can then be decoded back into pixel space for visualization, enabling action-conditioned video generation and long-term world simulation.

## 4    WorldPack

The design of WorldPack is motivated by a gap between two existing approaches to long-horizon conditioning. FramePack (Zhang & Agrawala, 2025) compresses past frames at varying rates to expand the effective context, but determines compression rates based on temporal proximity, frame similarity, or a hybrid of the two, which is not a direct proxy for 3D spatial relevance in world modeling. Spatial memory retrieval (Xiao et al., 2025; Yu et al., 2025a) selects the most relevant frames based on 3D co-visibility, but is constrained to a fixed context window and discards all non-selected frames. WorldPack unifies these ideas: it packs a large number of frames into the context while allocating compression rates based on spatial relevance (like memory retrieval), so that all historical frames are retained at a fidelity proportional to their importance. Algorithm 1 summarizes the complete procedure.

### 4.1    Video World Modeling with Conditional Diffusion Transformer

Following Section 3, we design $F_\theta$ as a probabilistic mapping to simulate stochastic environments. To this end, we employ CDiT (Bar et al., 2024), which is a temporally autoregressive transformer model, and where efficient CDiT blocks are applied $N$ times over the input sequence (Figure 1). Unlike a standard Transformer that applies self-attention across all tokens, CDiT restricts self-attention to the tokens of the denoised target frame and incorporates cross-attention over past frames, allowing efficient learning. This cross-attention contextualizes the representation through skip connections, and conditioning on input actions is incorporated. While a standard DiT (Peebles & Xie, 2023) can be directly applied, its computational complexity scales quadratically with context length, i.e., $O(m^2 n^2 d)$ for $n$ tokens per frame, $m$ frames, and token dimension $d$. In contrast, CDiT is dominated by the cross-attention complexity $O(mn^2 d)$, which scales linearly with context length, enabling the use of longer contexts.

In addition, our model must integrate memory contexts located at arbitrary temporal distances from the current timestep. To achieve this, we adopt Rotary Position Embeddings (RoPE) (Su et al., 2023) as

---

**Algorithm 1** WorldPack: Spatially-Aware Compressed Memory

---

**Require:** Historical frames $\{z_0, \ldots, z_t\}$, camera poses $\{p_0, \ldots, p_t\}$, current action $a_t$, context budget $L_{\text{pack}}$, uncompressed slots $S$
**Ensure:** Predicted next frame $z_{t+1}$

1: ▷ Spatial importance scoring
2: **for** each historical frame $z_i$ **do**
3:     $s_i \leftarrow \text{FoVOverlap}(p_i, p_t)$
4: **end for**
5: ▷ Geometric selection
6: Sort frames by $s_i$ in descending order
7: Assign top-$S$ frames to uncompressed slots ($d_i = 0$)
8: Assign the remaining frames to discrete compression groups according to their ranks
9: ▷ Trajectory packing
10: **for** each frame $z^i$ with priority $d_i$ **do**
11:     Encode at resolution $\ell_i = L_f / \lambda^{d_i}$
12: **end for**
13: Concatenate into packed context $\mathbf{z}_{\text{ctx}}$
14: ▷ Conditional generation via CDiT
15: $z_{t+1} \sim F_\theta(z_{t+1} \mid \mathbf{z}_{\text{ctx}}, a_t)$
16: **return** $z_{t+1}$

---

a position-aware design. RoPE enables consistent temporal representations regardless of variable context length, providing stable embeddings even for memory frames selected at arbitrary distances. This allows memory-aware inference over sequences with long-term dependencies.

## 4.2 Spatially-Aware Compressed Memory

Previous video world models are constrained by a fixed context length, preventing them from incorporating long-term history. While they remain sensitive to recent observations, predicting scenes that depend on events further in the past is challenging. This limitation causes errors to accumulate during rollouts, leading generated trajectories to gradually diverge from the original world (Decart et al., 2024; Guo et al., 2025).

To overcome this, we propose a spatially-aware compressed memory that combines hierarchical frame compression (i.e., trajectory packing) with 3D-guided rate allocation (i.e., geometric selection) into a single mechanism. Past frames are encoded at different resolutions depending on their spatial importance: frames that share a large field-of-view overlap with the current viewpoints are preserved at high resolution, while spatially less relevant frames are compressed and stored at lower resolution.

**Trajectory packing.** Let a sequence of frames selected from the historical trajectory be $z^0, z^1, \ldots, z^N$, where $N - 1$ is the number of frames maintained in the context window. After the Transformer patchifying process, each frame $z^i$ is assigned an effective context length $\ell_i$ determined by:

$$\ell_i = \frac{L_f}{\lambda^{d_i}}, \tag{4}$$

where $L_f$ is the base context length for high-resolution frames, $\lambda > 1$ controls compression intensity, and $d_i$ is the priority index of frame $z^i$. A lower $d_i$ indicates higher priority, resulting in more tokens and higher visual fidelity. The total packed context length is:

$$L_{\text{pack}} = \sum_{i=0}^{S-1} L_f + \sum_{i=S}^{N-1} \ell_i, \tag{5}$$

where $S$ denotes the number of uncompressed slots reserved for the most critical observations.

**Geometric selection.** The key question is how to assign $d_i$. FramePack (Zhang & Agrawala, 2025) assigns it based on temporal recency, frame similarity, or a hybrid of the two. However, temporal recency and generic frame similarity do not explicitly capture 3D spatial relevance: in world modeling, an agent revisiting a previously observed location may require high-fidelity access to a temporally distant frame even when its appearance differs due to viewpoint changes. We instead score each historical frame by how strongly it overlaps the current view in 3D space.

Each camera pose $p$ induces a truncated viewing frustum $V(p) \subset \mathbb{R}^3$ given the field-of-view angle and a near/far depth range. Following Xiao et al. (2025), we define the field-of-view overlap of a historical frame $i$ as

$$o_i = \frac{\text{Vol}\big(V(p_i) \cap V(p_t)\big)}{\text{Vol}\big(V(p_t)\big)} \in [0, 1], \tag{6}$$

estimated by Monte Carlo sampling of $M$ points in $V(p_t)$ and counting those also inside $V(p_i)$. To break ties among frames with similar overlap, where temporally closer frames tend to have less pose drift and fewer compounding generation artifacts, we add a mild temporal penalty:

$$s_i = w_o o_i - w_t \Delta t_i, \qquad \Delta t_i = \frac{t - i}{t}, \quad w_o > w_t > 0. \tag{7}$$

Frames are then sorted by $s_i$ and assigned compression: the top-$S$ go to the uncompressed slots ($d_i = 0$), and $d_i$ grows with rank for the rest. The contrast with temporal-proximity packing is that spatially critical frames are preserved at high resolution regardless of $\Delta t_i$. In all experiments, we estimate each FoV-overlap score using 10,000 Monte Carlo samples. We set $w_o = 1.0$ and $w_t = 0.2$, consistent with the settings used in WorldMem (Xiao et al., 2025). Because $\Delta t_i \in [0, 1]$, the temporal term acts primarily as a tie-breaking bias toward more recent frames when their FoV-overlap scores are similar.

**Implementation details.** Based on the rate-specific projection design of FramePack (Zhang & Agrawala, 2025), we use a discrete set of power-of-two compression rates and assign an independent input projection layer to each rate. We use three compression ratios $2^0, 2^2, 2^4$ ($\lambda = 2$ with $d_i \in \{0, 2, 4\}$). The packed context holds $S = 2$ uncompressed frames, 4 frames at ratio $2^2$, and 16 frames at ratio $2^4$, totaling $2 + 4 + 16 = 22$ historical frames. By Eq. 5 the packed length is $2L_f + (4/2^2)L_f + (16/2^4)L_f = 4L_f$, matching the budget of the 4-frame baseline while exposing $5.5\times$ more frames to the model. Each compression ratio uses an independent input projection layer, initialized by interpolating the pretrained patchify layer of the base model (kernel size $(4, 4)$). We retain the input configuration of NWM (Bar et al., 2024): images are resized to $224 \times 224$ and encoded using the Stable Diffusion VAE (Rombach et al., 2022), producing a $28 \times 28$ latent grid. The CDiT-B/2 backbone applies a $2 \times 2$ patch embedding, yielding $L_f = (28/2)^2 = 14 \times 14 = 196$ visual tokens per uncompressed frame.

## 5 Evaluation on Spatial Consistency

We primarily focus on evaluating video world models' ability to retain long-term spatial memory. For this purpose, we leverage LoopNav (Lian et al., 2025), a benchmark constructed in Minecraft environments. LoopNav is designed for loop-style navigation tasks, in which the agent explores a portion of the environment and then returns to an earlier location. This design provides a precise and targeted method for testing whether a model can recall and reconstruct previously observed scenes, making LoopNav a distinctive benchmark for evaluating spatial memory.

**Spatial Memory Retrieval Task (ABA).** The most basic setting of LoopNav is the A→B→A trajectory (Figure 2; **Left**). In this case, the segment from A to B acts as the exploration phase, supplying contextual observations to the model. The return path from B to A constitutes the reconstruction phase, during which the model must demonstrate spatial consistency in regenerating observations from earlier locations. Because the ground-truth sequence has already been observed, this scenario is best viewed as a spatial retrieval task that explicitly probes whether the model can reproduce information embedded in the context.

**Spatial Reasoning Task (ABCA).** Here, A→B→C forms the exploration phase, while C→A is evaluated as the reconstruction phase (Figure 2; **Right**). Unlike an A→B→A loop, this task challenges the model

Table 1: Model performance on tasks of varying type and difficulty. ABA denotes the spatial memory retrieval tasks, and ABCA denotes the spatial reasoning tasks. The navigation range (5, 15, 30, 50) indicates the size of the area within which the agent is required to move. SSIM (↑) evaluates better structural consistency, while LPIPS (↓) reflects perceptual fidelity, and both are reported as the mean and standard deviation over evaluation trajectories. We refer to baseline evaluation results from Lian et al. (2025).

| Nav. Range | Model | Context | Frames | SSIM ↑ | | LPIPS ↓ | |
|---|---|---|---|---|---|---|---|
| | | | | ABA | ABCA | ABA | ABCA |
| 5 | Oasis | 32 | 32 | $0.36_{\pm 0.13}$ | $0.34_{\pm 0.12}$ | $0.76_{\pm 0.09}$ | $0.82_{\pm 0.11}$ |
| | Mineworld | 32 | 32 | $0.31_{\pm 0.09}$ | $0.32_{\pm 0.10}$ | $0.73_{\pm 0.05}$ | $0.72_{\pm 0.07}$ |
| | DIAMOND | 32 | 32 | $\underline{0.40}_{\pm 0.10}$ | $\underline{0.37}_{\pm 0.09}$ | $0.75_{\pm 0.09}$ | $0.79_{\pm 0.09}$ |
| | NWM | 32 | 32 | $0.33_{\pm 0.11}$ | $0.31_{\pm 0.09}$ | $\underline{0.64}_{\pm 0.05}$ | $\underline{0.67}_{\pm 0.05}$ |
| | WorldPack (ours) | 4 | 22 | $\mathbf{0.41}_{\pm 0.13}$ | $\mathbf{0.44}_{\pm 0.22}$ | $\mathbf{0.50}_{\pm 0.08}$ | $\mathbf{0.48}_{\pm 0.19}$ |
| 15 | Oasis | 32 | 32 | $\underline{0.37}_{\pm 0.12}$ | $0.38_{\pm 0.14}$ | $0.82_{\pm 0.08}$ | $0.81_{\pm 0.10}$ |
| | Mineworld | 32 | 32 | $0.34_{\pm 0.13}$ | $0.32_{\pm 0.11}$ | $0.74_{\pm 0.08}$ | $0.74_{\pm 0.07}$ |
| | DIAMOND | 32 | 32 | $\mathbf{0.38}_{\pm 0.10}$ | $\underline{0.39}_{\pm 0.10}$ | $0.78_{\pm 0.08}$ | $0.79_{\pm 0.09}$ |
| | NWM | 32 | 32 | $0.30_{\pm 0.12}$ | $0.33_{\pm 0.12}$ | $\underline{0.67}_{\pm 0.03}$ | $\underline{0.65}_{\pm 0.05}$ |
| | WorldPack (ours) | 4 | 22 | $\mathbf{0.38}_{\pm 0.15}$ | $\mathbf{0.41}_{\pm 0.15}$ | $\mathbf{0.55}_{\pm 0.09}$ | $\mathbf{0.51}_{\pm 0.12}$ |
| 30 | Oasis | 32 | 32 | $\underline{0.33}_{\pm 0.11}$ | $\underline{0.35}_{\pm 0.11}$ | $0.86_{\pm 0.08}$ | $0.85_{\pm 0.09}$ |
| | Mineworld | 32 | 32 | $\underline{0.33}_{\pm 0.13}$ | $0.28_{\pm 0.09}$ | $0.77_{\pm 0.08}$ | $0.77_{\pm 0.08}$ |
| | DIAMOND | 32 | 32 | $\mathbf{0.37}_{\pm 0.10}$ | $\underline{0.35}_{\pm 0.10}$ | $0.81_{\pm 0.07}$ | $0.81_{\pm 0.08}$ |
| | NWM | 32 | 32 | $0.32_{\pm 0.11}$ | $0.30_{\pm 0.11}$ | $\underline{0.69}_{\pm 0.04}$ | $\underline{0.71}_{\pm 0.03}$ |
| | WorldPack (ours) | 4 | 22 | $\underline{0.33}_{\pm 0.14}$ | $\mathbf{0.37}_{\pm 0.18}$ | $\mathbf{0.62}_{\pm 0.08}$ | $\mathbf{0.57}_{\pm 0.12}$ |
| 50 | Oasis | 32 | 32 | $\underline{0.36}_{\pm 0.12}$ | $0.36_{\pm 0.11}$ | $0.86_{\pm 0.09}$ | $0.83_{\pm 0.07}$ |
| | Mineworld | 32 | 32 | $0.31_{\pm 0.16}$ | $0.32_{\pm 0.12}$ | $0.78_{\pm 0.12}$ | $0.75_{\pm 0.10}$ |
| | DIAMOND | 32 | 32 | $\mathbf{0.37}_{\pm 0.10}$ | $\mathbf{0.38}_{\pm 0.09}$ | $0.83_{\pm 0.09}$ | $0.81_{\pm 0.08}$ |
| | NWM | 32 | 32 | $0.28_{\pm 0.13}$ | $0.33_{\pm 0.11}$ | $\underline{0.72}_{\pm 0.08}$ | $\underline{0.65}_{\pm 0.04}$ |
| | WorldPack (ours) | 4 | 22 | $\underline{0.36}_{\pm 0.14}$ | $\underline{0.37}_{\pm 0.15}$ | $\mathbf{0.57}_{\pm 0.07}$ | $\mathbf{0.57}_{\pm 0.11}$ |

to rely on accumulated spatial memory to reconstruct the environment along an extended path, potentially across areas observed from different viewpoints or at earlier time steps. This setup is closely related to a spatial reasoning task, where success requires leveraging contextual knowledge to generate coherent future observations rather than simply retrieving frames.

**Metrics.** For evaluation, we use LPIPS (Zhang et al., 2018) to assess semantic-level perceptual fidelity, SSIM (Wang et al., 2004) to evaluate low-level structural alignment, and Fréchet Video Distance (FVD) (Unterthiner et al., 2019) to evaluate video synthesis quality. We further employ DreamSim (Fu et al., 2023), which measures perceptual similarity based on deep feature representations, and PSNR to capture pixel-level reconstruction quality. Since no single metric fully reflects semantic accuracy or long-term spatial coherence, we complement these quantitative results with qualitative inspection by human observers.

# 6 Experiments

## 6.1 Baselines

Oasis (Decart et al., 2024) is a world model that employs a ViT (Dosovitskiy et al., 2020) as a spatial autoencoder and a

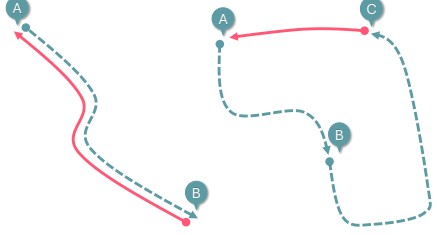

Spatial Retrieval Task    Spatial Reasoning Task

Figure 2: Illustration of the two LoopNav benchmark tasks. (**Left**) Spatial Memory Retrieval Task: the agent explores along A→B (blue path) and must reconstruct earlier observations on the return path B→A (red path). (**Right**) Spatial Reasoning Task: the agent explores along A→B→C (blue path) and must reconstruct the environment on the longer return path C→A (red path), requiring reasoning across accumulated spatial memory.

Table 2: Evaluation of models on spatial memory (ABA) and reasoning (ABCA) tasks under different navigation ranges. PSNR (↑) reflects pixel-level reconstruction accuracy, DreamSim (↓) captures deep-feature-based perceptual similarity, and both are reported as the mean and standard deviation over evaluation trajectories, whereas FVD (↓) measures temporal video quality as a distributional distance computed over the entire set of evaluation videos. † Our implementation.

| Nav. Range | Model | Context | Frames | PSNR ↑ | | DreamSim ↓ | | FVD ↓ | |
|---|---|---|---|---|---|---|---|---|---|
| | | | | ABA | ABCA | ABA | ABCA | ABA | ABCA |
| 5 | NWM† | 4 | 4 | $12.2_{\pm1.7}$ | $13.5_{\pm5.8}$ | $0.31_{\pm0.05}$ | $0.31_{\pm0.17}$ | 1847 | **1997** |
| | WorldPack (ours) | 4 | 22 | $\mathbf{13.3}_{\pm1.8}$ | $\mathbf{14.6}_{\pm5.2}$ | $\mathbf{0.27}_{\pm0.07}$ | $\mathbf{0.26}_{\pm0.13}$ | **1510** | 2004 |
| 15 | NWM† | 4 | 4 | $11.2_{\pm1.4}$ | $12.2_{\pm4.2}$ | $0.42_{\pm0.09}$ | $0.37_{\pm0.17}$ | 2114 | 1789 |
| | WorldPack (ours) | 4 | 22 | $\mathbf{12.7}_{\pm2.0}$ | $\mathbf{13.7}_{\pm3.5}$ | $\mathbf{0.32}_{\pm0.09}$ | $\mathbf{0.27}_{\pm0.08}$ | **1449** | **1339** |
| 30 | NWM† | 4 | 4 | $10.4_{\pm2.0}$ | $11.3_{\pm2.3}$ | $0.47_{\pm0.12}$ | $0.38_{\pm0.16}$ | 1992 | 2175 |
| | WorldPack (ours) | 4 | 22 | $\mathbf{11.2}_{\pm2.4}$ | $\mathbf{11.9}_{\pm2.0}$ | $\mathbf{0.42}_{\pm0.08}$ | $\mathbf{0.35}_{\pm0.11}$ | **1777** | **1619** |
| 50 | NWM† | 4 | 4 | $8.7_{\pm2.3}$ | $10.8_{\pm2.2}$ | $0.52_{\pm0.17}$ | $0.43_{\pm0.19}$ | 2121 | 1983 |
| | WorldPack (ours) | 4 | 22 | $\mathbf{11.9}_{\pm1.8}$ | $\mathbf{12.1}_{\pm1.9}$ | $\mathbf{0.35}_{\pm0.07}$ | $\mathbf{0.34}_{\pm0.09}$ | **1624** | **1440** |

DiT (Peebles & Xie, 2023) as the latent diffusion backbone, trained with Diffusion Forcing (Chen et al., 2024). It generates frames autoregressively with user-controllable conditioning, and the publicly available Oasis-500M model is evaluated with a context length of 32. Mineworld (Guo et al., 2025) is an interactive world model based on a pure Transformer architecture that generates new scenes from paired game frames and actions, with its pretrained checkpoint evaluated at a context length of 32. DIAMOND (Alonso et al., 2024) is a diffusion-based world model built upon a UNet architecture (Ronneberger et al., 2015), generating frames conditioned on past observations and actions, and evaluated with a context length of 32. NWM (Bar et al., 2024) is a controllable video generation model that predicts future observations conditioned on navigation actions, leveraging CDiT with a context length of 32 or 4.

## 6.2 Results

In the multi-step rollout generation (Table 1 and Table 2), WorldPack, despite the shortest context length, generally outperforms the baselines – Oasis, Mineworld, DIAMOND, and NWM – in SSIM and LPIPS, and also surpasses NWM in PSNR, DreamSim, and FVD. However, the SSIM results were not decisively superior, remaining only partially competitive. This tendency can be explained by the inherent limitations of distortion-based metrics, which favor spatially averaged or blurred predictions that minimize pixel-wise differences at the expense of perceptual fidelity (Blau & Michaeli, 2018). Indeed, Lian et al. (2025) also reported that SSIM exhibits only a weak correlation with perceptual quality in visualizations.

Collectively, these results demonstrate consistent improvements across both the ABA and ABCA tasks, as evidenced by quantitative metrics across all navigation ranges. In particular, the proposed compressed memory mechanism plays a crucial role in balancing high context efficiency with long-term spatial consistency. Accommodating more frames than uncompressed baselines allows the essential frames for world modeling to remain accessible even under the shortest context-length constraints.

## 6.3 Ablation Study

To evaluate the individual contributions of trajectory packing and geometric selection, we conducted an ablation study comparing the following four configurations. For a fair comparison, all settings are constrained to a fixed context size of four frames.

- **Baseline:** Following the standard approach (Bar et al., 2024), the four most recent frames are used directly as the context.
- **Nearest Frame Packing:** Following the protocol in FramePack (Zhang & Agrawala, 2025), the 22 most recent frames are compressed into a 4-frame context, with compression rates determined by temporal proximity.

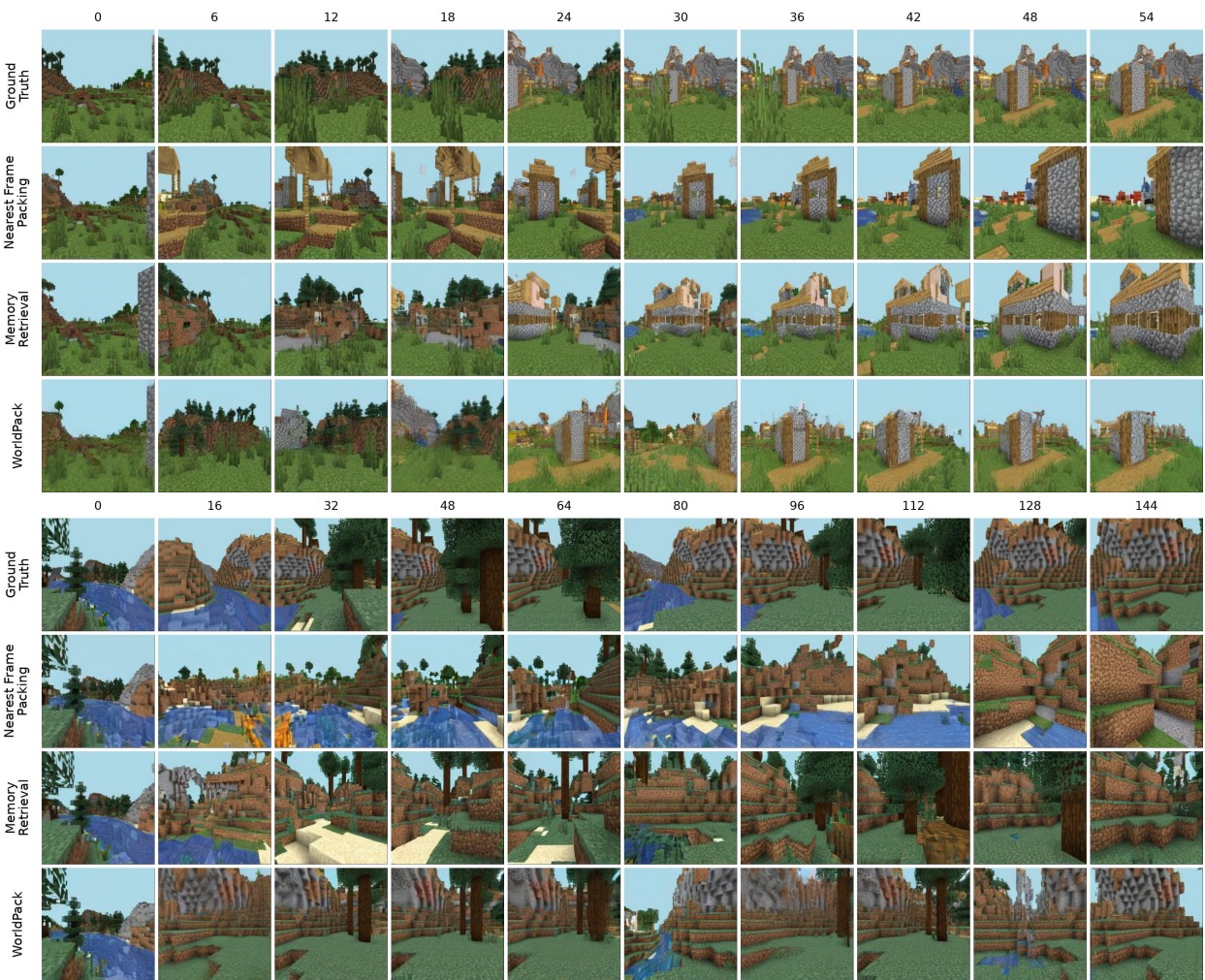

Figure 3: Qualitative comparison of rollouts. We compare Ground Truth, Nearest Frame Packing, Memory Retrieval, and WorldPack. WorldPack preserves the environment's spatial structure more consistently than the other variants.

- **Memory Retrieval:** Following the spatial memory retrieval mechanism of WorldMem (Xiao et al., 2025) and Context-as-Memory (Yu et al., 2025a), the context consists of the four frames with the highest FoV-based spatial similarity scores. This setting serves as a direct comparison with these retrieval-based methods.

- **WorldPack (ours):** The 22 frames are compressed into a 4-frame context, where compression rates are determined based on spatial similarity scores.

First, the results of the ablation study for ABA-5 in LoopNav are presented in Table 3. The comparison between Nearest Frame Packing and WorldPack demonstrates the effectiveness of geometric selection, which adaptively determines compression rates based on 3D-aware importance rather than employing a fixed rate. Furthermore, since the Memory Retrieval setting reproduces the selection mechanism of WorldMem (Xiao et al., 2025) and Context-as-Memory (Yu et al., 2025a), its comparison with WorldPack serves as a direct evaluation against these retrieval-based methods and highlights the efficacy of trajectory packing, which enables the handling of larger frame sizes without increasing context length by compressing past frame information. Figure 3 qualitatively compares the three packing/retrieval variants against the ground truth: WorldPack preserves the environment's spatial structure more faithfully than Nearest Frame Packing and

Table 3: Ablation study of WorldPack on ABA-5 in LoopNav. Each baseline ablates one component: Nearest Frame Packing applies trajectory packing without geometric selection (TP only), while Memory Retrieval applies geometric selection without trajectory packing (GS only). Thus, the gap between Nearest Frame Packing and WorldPack isolates the effect of geometric selection (GS), and the gap between Memory Retrieval and WorldPack isolates the effect of trajectory packing (TP). DreamSim, LPIPS, PSNR, and SSIM are reported as the mean and standard deviation over evaluation trajectories, whereas FVD is computed as a distributional distance over the entire set of evaluation videos.

| Method | TP | GS | DreamSim ↓ | LPIPS ↓ | PSNR ↑ | SSIM ↑ | FVD ↓ |
|---|---|---|---|---|---|---|---|
| Baseline | ✗ | ✗ | $0.31_{\pm 0.05}$ | $0.53_{\pm 0.08}$ | $12.2_{\pm 1.7}$ | $0.39_{\pm 0.14}$ | 1847 |
| Nearest Frame Packing | ✓ | ✗ | $0.28_{\pm 0.05}$ | $0.51_{\pm 0.09}$ | $13.0_{\pm 1.1}$ | $\mathbf{0.42}_{\pm 0.14}$ | 1683 |
| Memory Retrieval | ✗ | ✓ | $0.30_{\pm 0.07}$ | $0.51_{\pm 0.09}$ | $12.8_{\pm 1.7}$ | $0.41_{\pm 0.14}$ | 1694 |
| WorldPack (ours) | ✓ | ✓ | $\mathbf{0.27}_{\pm 0.07}$ | $\mathbf{0.50}_{\pm 0.08}$ | $\mathbf{13.3}_{\pm 1.8}$ | $0.41_{\pm 0.13}$ | **1510** |

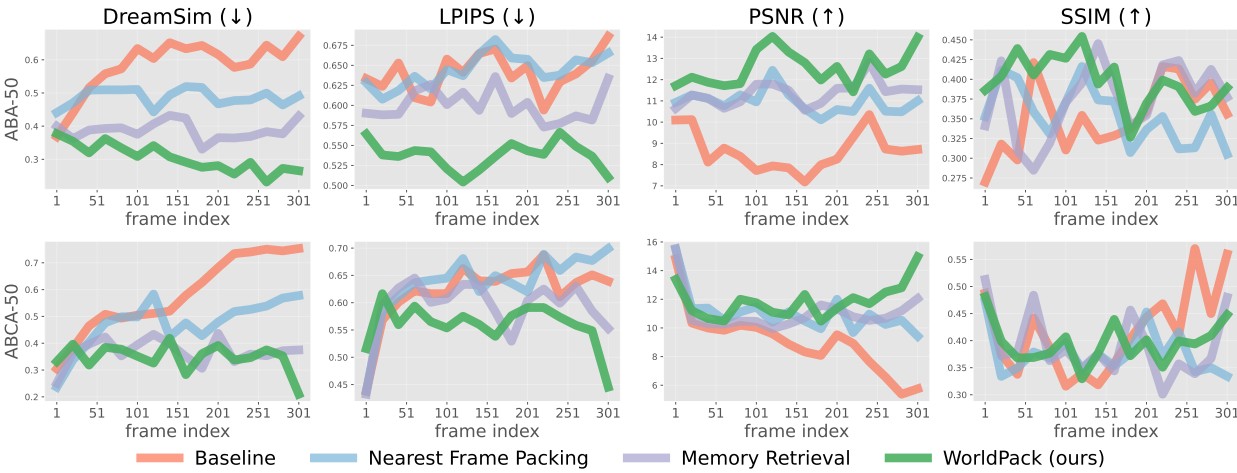

Figure 4: Prediction performance on the terminal frames of trajectories with different navigation ranges. **Top**: 301 frames rollout in ABA-50. **Bottom**: 301 frames rollout in ABCA-50. WorldPack not only accesses task-relevant information based on 3D spatial cues but also retains a significantly larger number of frames within the context through frame compression. Consequently, the model can effectively correct the generation by fully leveraging past observations, thereby minimizing quality degradation.

Memory Retrieval across the displayed frames. Collectively, these results suggest that both components are vital for robust world modeling.

Next, for a more detailed analysis, Figure 4 illustrates the transitions of each metric throughout a 301-frame rollout for the LoopNav ABA-50 and ABCA-50 tasks. Nearest Frame Packing sometimes shows performance improvements during the initial stages of the rollout, as it can maintain a larger context and allow for longer access to past observations (e.g., in ABCA-50). However, as generations progress, past observations are eventually evicted from the context window, leading to a gradual degradation in generation quality. Memory Retrieval, which corresponds to the retrieval mechanism of WorldMem (Xiao et al., 2025) and Context-as-Memory (Yu et al., 2025a), can extract past information essential for prediction based on 3D spatial proximity scoring. While this helps mitigate the divergence in generation quality to some extent, its effectiveness is limited by the fixed context length, which restricts the total number of frames the model can handle simultaneously. In contrast, WorldPack not only accesses task-relevant information based on 3D spatial cues but also retains a significantly larger number of frames within the context through frame compression. Consequently, the model can effectively correct the generation by fully leveraging past observations, thereby minimizing quality degradation. This advantage is particularly evident in the latter segments of ABCA-

Table 4: Evaluation on RECON dataset, real-world generation performance, including DreamSim (↓), LPIPS (↓), PSNR (↑), SSIM (↑), and FVD (↓).

| Model | Context | Frames | DreamSim ↓ | LPIPS ↓ | PSNR ↑ | SSIM ↑ | FVD ↓ |
|---|---|---|---|---|---|---|---|
| Baseline | 4 | 4 | $0.25_{\pm0.11}$ | $0.48_{\pm0.08}$ | $12.9_{\pm2.4}$ | $0.36_{\pm0.11}$ | 822 |
| WorldPack | 4 | 22 | $\mathbf{0.17}_{\pm0.05}$ | $\mathbf{0.44}_{\pm0.07}$ | $\mathbf{13.7}_{\pm2.4}$ | $\mathbf{0.41}_{\pm0.12}$ | **694** |

Table 5: Single-step inference time and memory usage of the diffusion model. The reported inference times exclude the computation of FoV overlap scores used for geometric selection.

| Model | Frames | Inference Time (1-step, sec) | Memory Usage (GB) |
|---|---|---|---|
| Baseline | 4 | 0.255 | 22.7 |
| WorldPack | 22 | 0.296 | 25.4 |

Table 6: Runtime of FoV-based geometric selection performed on a single NVIDIA H100.

| Memory Candidates | Selection Time (sec) |
|---|---|
| 50 | 0.05 |
| 100 | 0.06 |
| 400 | 0.10 |
| 1600 | 0.26 |

50, a spatial reasoning task, where WorldPack demonstrates significant performance gains. In such spatial reasoning tasks, the importance of past observations for accurate prediction is maximized during the latter stages of the rollout (see Figure 2; **Right**). While WorldPack successfully leverages this information to improve generation, other methods fail to recover quality, either because they cannot access past observations or lack sufficient context capacity to retain them.

### 6.4 Experiments with Real-World Data

To verify the practical usefulness of WorldPack beyond simulator environments such as Minecraft, we conducted experiments using real-world data. Specifically, using NWM (Bar et al., 2024) as the base model, we evaluated our method on the RECON dataset (Shah et al., 2021), one of the most commonly used datasets in prior video-generation world-model studies (Shah et al., 2022; Sridhar et al., 2024; Bar et al., 2024). In our experiments, we used the first 80 frames as context and generated the subsequent frames. The quantitative results are shown in Table 4. These results demonstrate that WorldPack achieves strong generative performance even on real-world data, confirming its effectiveness beyond simulated environments.

### 6.5 Analysis of Computational Efficiency

We separately evaluate the computational costs of diffusion-model inference and FoV-based geometric selection. Table 5 reports the single-step inference time and memory usage of the diffusion model, excluding the computation of FoV overlap scores. Compared with the baseline, WorldPack increases the number of visible historical frames from 4 to 22, corresponding to a 5.5× increase, while increasing diffusion-model inference time by approximately 16% and memory usage by approximately 12%. These results show that trajectory packing substantially expands the accessible history with moderate computational and memory overhead.

FoV-based geometric selection introduces an additional retrieval cost that depends on the number of memory candidates. As shown in Table 6, retrieval takes 0.05 s for 50 candidates, 0.06 s for 100 candidates, 0.10 s for 400 candidates, and 0.26 s for 1,600 candidates. The runtime scales approximately linearly with the number of candidates, indicating predictable computational growth as the accessible context is expanded. Thus, the additional cost remains limited for hundreds of memory candidates, although it increases as the trajectory history becomes longer.

# 7 Discussion and Limitation

In this study, we focused on memory management for world modeling and employed a 3D scoring mechanism based on camera poses—a widely adopted approach in existing literature—to determine frame importance (HunyuanWorld, 2025; Yu et al., 2025a; Xiao et al., 2025). However, it has been noted that such scoring methods, which rely heavily on 3D information, may underperform in complex environments with occlusion (Xiao et al., 2025). Consequently, exploring more robust scoring metrics that can overcome these constraints will be crucial for achieving more sophisticated and reliable world modeling in the future. In addition, we primarily focused on the simulation capabilities of video world models and therefore evaluated their scene-generation performance. Another practical limitation is that FoV overlap assumes camera poses for the current and historical frames. In real-world settings, these poses may be estimated using standard SLAM or visual-inertial odometry systems (Campos et al., 2021; Qin et al., 2018; Teed & Deng, 2022), although their pose errors may affect frame prioritization. An important direction for future work is therefore to develop selection mechanisms that explicitly account for pose uncertainty and to evaluate their robustness under realistic localization noise. As a future direction, we believe that exploring policy learning and planning with video world models (Alonso et al., 2024) will further deepen the discussion on the utility of spatial memory capabilities.

# 8 Conclusion

We introduced WorldPack, a video world model that achieves long-horizon spatial consistency through spatially-aware compressed memory. By unifying trajectory packing with geometric selection, WorldPack retains substantially more historical context than prior methods while preserving high fidelity for the frames most relevant to spatial reasoning. Experiments on LoopNav and RECON reveal that simply expanding context length is less effective than intelligently compressing a larger history with spatial guidance and spatially adaptive compression rates, which provide clear benefits over both temporal-proximity-based packing and fixed-context spatial retrieval, with the advantage growing over the rollout horizon. Additionally, a controlled comparison that reproduces WorldMem's retrieval mechanism within our backbone confirms that the contribution lies not in spatial scoring itself but in using it to control compression rates across a larger set of frames.

## Broader Impact Statement

This work studies memory mechanisms for video world models. Potential positive impacts include more efficient simulation and planning systems. Potential risks include the misuse of increasingly realistic interactive video generation systems to create deceptive or harmful content. Our work does not introduce new data collection or human-subject experiments, and our experiments are conducted on public navigation benchmarks. We encourage future deployments to incorporate provenance, access control, and safety evaluation.

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

# Appendix

## A  The Use of Large Language Models

In this paper, we mainly used LLMs to polish writing and propose paraphrases.

## B  Evaluation Metrics

To evaluate the perceptual consistency of generated outputs, we use Learned Perceptual Image Patch Similarity (LPIPS) (Zhang et al., 2018) and DreamSim (Fu et al., 2023). These metrics measure perceptual similarity between generated and ground-truth images using deep features extracted from neural networks. Following Lian et al. (2025), we use VGG as the backbone for LPIPS. We additionally use PSNR and SSIM to evaluate pixel-level reconstruction quality.

To evaluate video generation quality, we use Fréchet Video Distance (FVD) (Unterthiner et al., 2019). FVD is a set-level metric that computes the Fréchet distance between the feature distributions of real and generated video sets, using features extracted by I3D (Carreira & Zisserman, 2018). Lower values indicate that the distribution of generated videos is closer to that of real videos. The public LoopNav implementation computes FVD separately for each pair of real and generated videos and then averages the resulting scores across evaluation trajectories[1]. Because FVD was originally defined as a distributional distance between sets of videos, we instead compute FVD jointly over all evaluation videos within each task–range setting. LPIPS, DreamSim, PSNR, and SSIM are reported as the mean and standard deviation across evaluation trajectories, whereas FVD is reported as a single distributional distance computed over the entire evaluation video set.

## C  Dataset and Evaluation Details

### C.1  LoopNav

As described in Section 5, we adopt LoopNav (Lian et al., 2025) as the primary benchmark for evaluating long-term spatial memory in video world models. LoopNav uses loop-based navigation trajectories in which an agent revisits previously observed regions, thereby enabling a structured evaluation of whether a model can retain temporally distant observations and reuse them during generation upon revisitation. Specifically, the ABA task evaluates the retrieval and reconstruction of previously observed scenes, whereas the ABCA task evaluates the ability to leverage spatial memory accumulated over a longer trajectory.

---

[1] `https://github.com/Kevin-lkw/LoopNav/blob/main/baseline/metrics/calcmetric.py` (accessed July 17, 2026).

| Benchmark | # Evaluation videos | Max. video length | # Total frames |
|---|---|---|---|
| MineWorld (Guo et al., 2025) | 1,000 | 16 | 16,000 |
| WorldMem (Xiao et al., 2025) | 300 | 100 | 30,000 |
| LoopNav (Lian et al., 2025) | 144 | 630 | 30,738 |

Table 7: Comparison of evaluation scale across video world-modeling benchmarks. Although LoopNav uses fewer evaluation videos than MineWorld and WorldMem, it includes longer rollouts and a comparable total number of frames to WorldMem, enabling a more structured evaluation of long-term spatial memory in video world models.

**Dataset and training split.** LoopNav is constructed from Minecraft navigation trajectories and evaluates whether a video world model can preserve spatial consistency when an agent revisits previously observed regions. We follow the official train–test split and ensure that trajectories used for evaluation do not appear in the training set. For training, we use 17,280 trajectories.

**Evaluation tasks.** We evaluate two types of loop-navigation trajectories. The ABA setting evaluates spatial memory retrieval, in which the agent follows an $A \to B \to A$ trajectory and is evaluated while returning to a previously observed region. The ABCA setting evaluates spatial reasoning over a longer $A \to B \to C \to A$ trajectory. For both tasks, we consider four navigation ranges: $\{5, 15, 30, 50\}$. Following evaluation settings in Lian et al. (2025), we evaluate 18 trajectories, for each combination of task type and navigation range. The main LoopNav evaluation therefore contains $18 \times 4 \times 2 = 144$ condition-specific evaluation rollouts in total. To clarify the scale of our evaluation, we further compare LoopNav with benchmarks used in related work. Table 7 reports the number of evaluation videos, rollout length, and total number of frames for MineWorld (Guo et al., 2025), WorldMem (Xiao et al., 2025), and LoopNav evaluation. Although our LoopNav evaluation contains fewer videos than MineWorld and WorldMem, its total number of generated frames is comparable to that of WorldMem and includes substantially longer rollout scenarios, with trajectories of up to 630 generated frames. Thus, while the number of trajectories and trajectory length represent complementary aspects of evaluation, our setting is well aligned with the objective of assessing the long-term spatial memory targeted by WorldPack.

## C.2 RECON

RECON (Shah et al., 2021) contains robot navigation trajectories collected across diverse real-world environments, and we use it to evaluate whether WorldPack's long-term memory capabilities generalize beyond simulated settings. We use all 9,468 trajectories in the training split. During evaluation, we provide the first 80 frames of each trajectory as the initial observation sequence and autoregressively generate the subsequent frames, allowing us to assess whether the model can exploit a sufficiently long history during future prediction. Accordingly, we evaluate only trajectories containing more than 80 frames, resulting in a total of 15 out of 64 total rollout evaluation trajectories.

# D   Training Details

This appendix provides additional details on the training setup and hyperparameters used in our experiments. We summarize the common training and diffusion settings in Table 8, and the configuration of each variant in Table 9 and Table 10. Following NWM (Bar et al., 2024), we use latent-space diffusion with the Stable Diffusion VAE (Rombach et al., 2022) and the same basic image preprocessing and diffusion-model setup. Unless otherwise specified, experiments are conducted on four NVIDIA H100 GPUs.

Table 8: Training and diffusion settings used across our experiments.

| Item | Setting |
|---|---|
| Backbone | CDiT-B/2 (Bar et al., 2024) |
| Image resolution | $224 \times 224$ |
| VAE | `stabilityai/sd-vae-ft-ema` |
| Latent scaling factor | 0.18215 |
| Diffusion timesteps | 1000 |
| Noise schedule | Linear |
| Optimizer | AdamW (Loshchilov & Hutter, 2019) |
| Learning rate | $8 \times 10^{-5}$ |
| Weight decay | 0 |
| Learning-rate schedule | Constant |
| EMA decay | 0.9999 |
| Evaluation weights | EMA weights |
| Mixed precision | bfloat16 |
| Gradient clipping | 10.0 |
| Batch size | 8 |

Table 9: Training and ablation configuration for LoopNav dataset. The baseline is trained from scratch, while the other variants are fine-tuned from the baseline.

|  | Baseline | Nearest Frame Packing | Memory Retrieval | WorldPack |
|---|---|---|---|---|
| Training steps | 1,000,000 | +200,000 | +200,000 | +200,000 |
| Visible frames | 4 | 22 | 4 | 22 |
| Context budget | 4 | 4 | 4 | 4 |

Table 10: Training and ablation configuration for RECON dataset. The baseline is trained from scratch, while WorldPack is fine-tuned from the baseline.

|  | Baseline | WorldPack |
|---|---|---|
| Training epochs | 300 | +300 |
| Visible frames | 4 | 22 |
| Context budget | 4 | 4 |

# E   Further Ablation Study

**Encoding Spatial Information Helps World Modeling.** We investigate the impact of encoding spatial information on world modeling. Following Sitzmann et al. (2021); Xiao et al. (2025), we adopt Plücker embedding to convert 5D poses $p \in \mathbb{R}^5$ into dense positional features $PE(p) \in \mathbb{R}^{h \times w \times 6}$, consistent with recent works (He et al., 2024; Gao et al., 2024a). As shown in Table 11, removing the camera pose embedding (`w/o Camera Pose Embedding`) results in performance degradation across key metrics, including DreamSim and LPIPS. These results confirm that explicitly injecting spatial information via camera poses is highly effective for enhancing the understanding of 3D structures and improving prediction accuracy in memory-based world modeling.

**Too Much Compression Collapses World Modeling.** Next, we examine the effect of compression rates in the tokenizer on model performance. While our main method employs a frame-wise tokenizer with packing limited to the spatial dimension, this ablation study investigates configurations that incorporate temporal compression (Table 12).

First, we observed that compressing only the temporal dimension (`+ Temporal Compression`) improves performance compared to the baseline. This improvement is likely due to temporal compression, which allows the model to handle longer frame sequences within the same token budget, enabling the world model to leverage a broader range of past information. However, when further spatial compression (`+ Nearest Frame Packing`) or spatio-temporal compression (`+ Temporal Packing`) is applied, the performance performance may not improve and may even deteriorate. These findings suggest that excessive compression leads to significant information loss, which outweighs the benefits of an extended context length. This confirms a critical trade-off between representation density and context length in effective world modeling.

Table 11: Ablation for encoding spatial information.

| Method | DreamSim ↓ | LPIPS ↓ | PSNR ↑ | SSIM ↑ | FVD ↓ |
|---|---|---|---|---|---|
| Baseline | 0.31 ±0.05 | 0.53 ±0.08 | 12.2 ±1.7 | 0.39 ±0.14 | 1847 |
| Memory Retrieval | 0.30 ±0.07 | 0.51 ±0.09 | 12.8 ±1.7 | 0.41 ±0.14 | 1694 |
| w/o Camera Pose Embedding | 0.31 ±0.08 | 0.51 ±0.08 | 12.7 ±1.9 | 0.40 ±0.13 | 2067 |

Table 12: Ablation for compression rate and world modeling performance

| Method | Context | Frames | DreamSim ↓ | LPIPS ↓ | PSNR ↑ | SSIM ↑ | FVD ↓ |
|---|---|---|---|---|---|---|---|
| Baseline | 4 | 4 | 0.31 ±0.05 | 0.53 ±0.08 | 12.2 ±1.7 | 0.39 ±0.14 | 1847 |
| + Temporal Compression | 4 | 16 | 0.29 ±0.05 | 0.51 ±0.08 | 12.5 ±1.5 | 0.40 ±0.13 | 1774 |
| + Nearest Frame Packing | 4 | 88 | 0.30 ±0.09 | 0.51 ±0.09 | 13.1 ±2.4 | 0.43 ±0.16 | 1714 |
| + Temporal Packing | 4 | 296 | 0.32 ±0.06 | 0.53 ±0.09 | 12.6 ±1.7 | 0.37 ±0.14 | 1899 |

## F Prediction Performance for Rollout

We describe LoopNav rollout results for ABA-{5, 15} and ABCA-{5, 15} in Figure 5, and for ABA-{30, 50} and ABCA-{30, 50} in Figure 6.

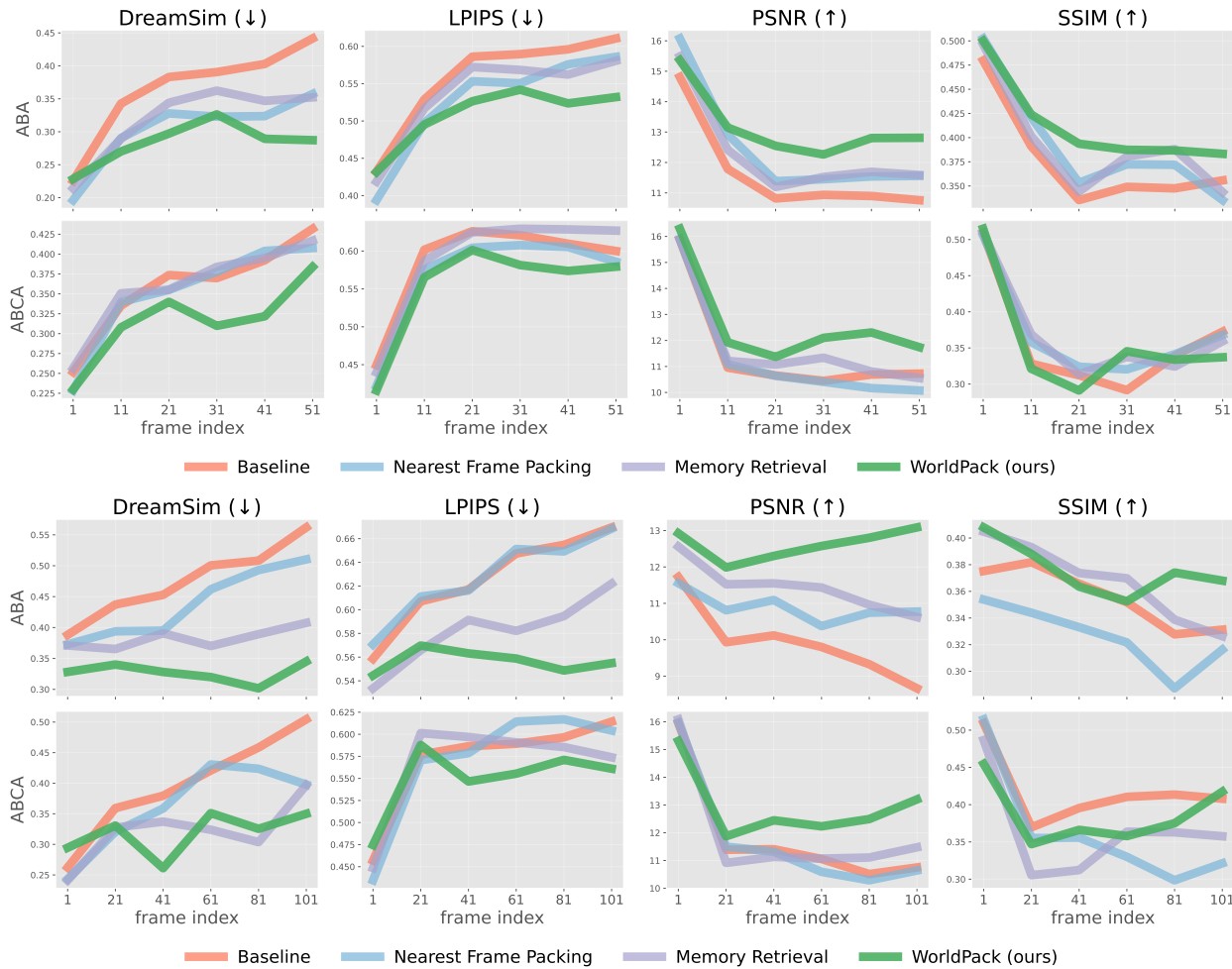

Figure 5: Prediction performance on the terminal frames of ABCA trajectories with different navigation ranges. **Top**: last 51 frames in ABA-5 and ABCA-5. **Bottom**: last 101 frames in ABA-15 and ABCA-15. WorldPack not only accesses task-relevant information based on 3D spatial cues but also retains a significantly larger number of frames within the context through frame compression. Consequently, the model can effectively correct the generation by fully leveraging past observations, thereby minimizing quality degradation.

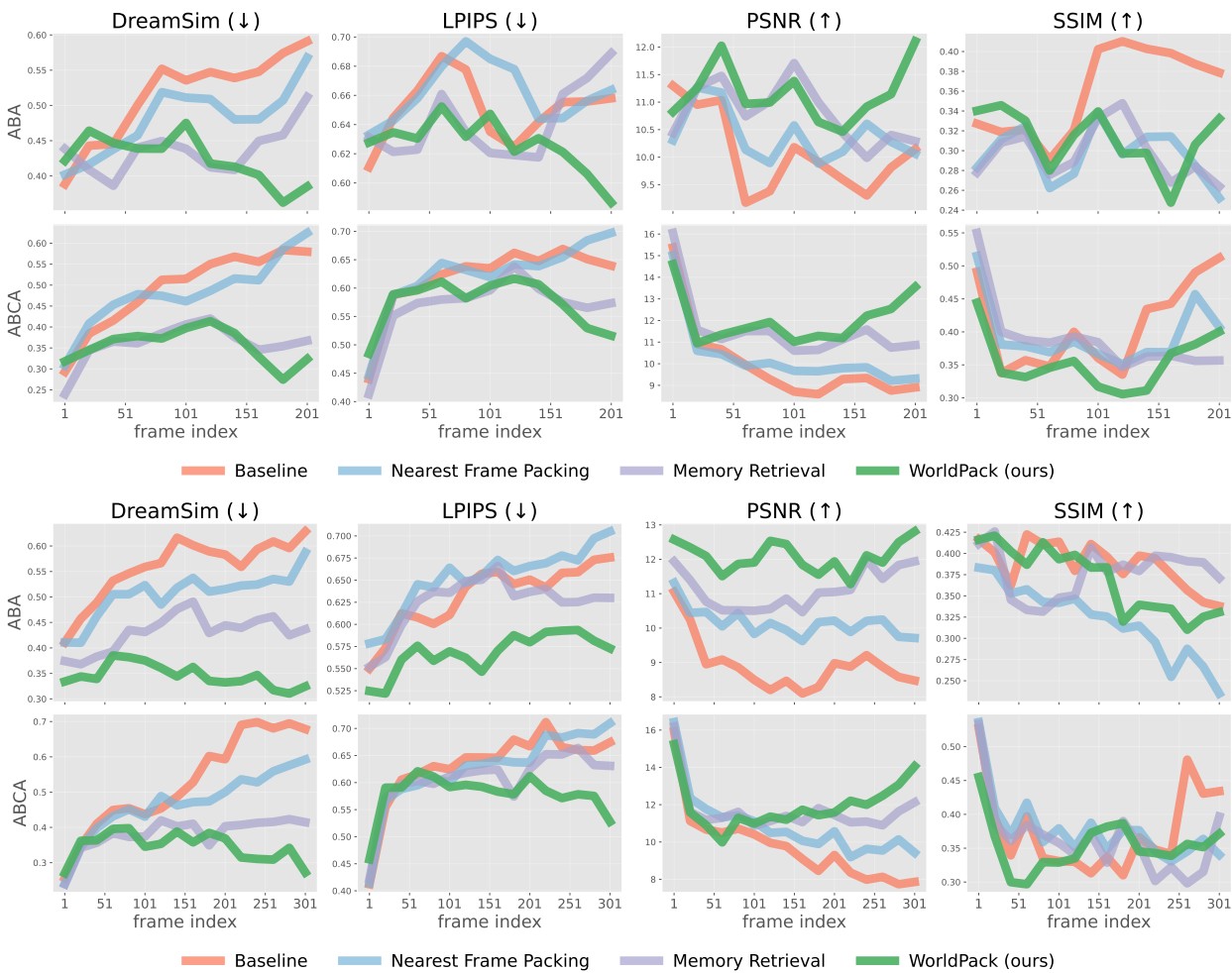

Figure 6: Prediction performance on the terminal frames of ABCA trajectories with different navigation ranges. **Top**: last 201 frames in ABA-30 and ABCA-30. **Bottom**: last 301 frames in ABA-50 and ABCA-50. WorldPack not only accesses task-relevant information based on 3D spatial cues but also retains a significantly larger number of frames within the context through frame compression. Consequently, the model can effectively correct the generation by fully leveraging past observations, thereby minimizing quality degradation.

## G  FVD for Full Trajectories

As discussed in Appendix C.1, our evaluation on LoopNav uses 144 trajectories. In the main text, FVD is computed separately for each task type and navigation range, and thus each reported FVD value is based on 18 trajectories. To mitigate potential concerns regarding the limited number of videos in each condition, we additionally compute FVD jointly over all 144 evaluation trajectories. As shown in Table 13, WorldPack achieves the lowest FVD, outperforming Memory Retrieval, Nearest Frame Packing, and the baseline. This result confirms that the improvement of WorldPack remains consistent when FVD is computed over a larger evaluation set.

Table 13: FVD computed jointly over 144 LoopNav evaluation trajectories. Nearest Frame Packing uses trajectory packing without geometric selection (TP only), while Memory Retrieval uses geometric selection without trajectory packing (GS only). Lower is better.

| Method | TP | GS | FVD ↓ |
|---|---|---|---|
| Baseline | ✗ | ✗ | 872 |
| Nearest Frame Packing | ✓ | ✗ | 846 |
| Memory Retrieval | ✗ | ✓ | 676 |
| WorldPack (ours) | ✓ | ✓ | **608** |

