# OpenReview forum: "WorldPack: Dynamic Frame Compression for Long-context Video World Modeling"
_TMLR — Under review for TMLR_

### Review · Reviewer_DEFy · 2026-06-20

**Summary Of Contributions:**

The manuscript proposes a video world model that introduces spatially-aware compressed memory. The approach is built on packing historical frames into a fixed-length context and dynamically allocating compression rates based on 3D spatial relevance. The contributions are, in my view, (1) proposing using spatial relevance (instead of temporal proximity) on a world model built on FramePack and (2) experiments suggesting the mechanism helps.

**Audience:**

Yes

**Audience Explanation:**

Video world modeling is an active research topic and the proposed methodology describes a system (and accompanying empirical results) that was not reported in the literature.

The combination of frame packing with dynamic compression rates based on spatial relevance is (arguably) a straightforward synthesis of prior work, but there is value for the community in knowing the proposed system works.

**Broader Impact Concerns:**

I have no broader impact concerns that would require further discussion.

**Claims And Evidence:**

No

**Claims Explanation:**

There are three significant limitations of the work.

1. The predictive power of the main experiment is very limited (the main experiment appears to report numbers only on the 18 navigation tasks that LoopNav reports numbers on).
2. The LoopNav benchmark itself has minimal external adoption (no peer reviewed work that I know uses it), which makes it hard to assess how informative the empirical results are.
3. Standard deviation is omitted, only mean is reported.

Given that the empirical evaluation is a big part of the contribution, the fact that a niche benchmark with a small evaluation set is the main empirical validation of the approach greatly diminishes the predictive power of the main experiment.

**Requested Changes:**

Critical changes necessary for the claims to be supported by evidence:

1. The reported results completely omit statistics beyond the aggregates (e.g., standard deviation across the eval set). E.g., LoopNav reports everything as mean +- stdev. I strongly encourage authors to not drop stdev, especially given that empirical results are a big part of the contribution.w

2. Please explicitly state the number of evaluation trajectories in all experiments. If you use only 18 in the main experiment, please use more of the test set (there appears to be 960 test trajectories available).
	- Other benchmarks have more than an order of magnitude larger evaluation sets (e.g., Mineworld 1000 test clips, WorldMem 300 test videos). Note that in video world modelling, the fact that frames within a video are highly correlated makes it necessary to increase trajectory sample size, rather than only increasing the frame-level sample size (i.e., more videos is valuable and in that sense orthogonal to longer videos).
	- If you use only the 18 eval trajectories mentioned above in the main experiment, this exacerbates the lack of standard deviation reporting. Furthermore, FVD might be a poor choice at this sample size.
	- The use of RECON does not solve this problem, because (1) the sample size is not reported, and (2) that experiment is itself more limited as it only compares with a single baseline.

3. Explicitly acknowledge the limitations of the experiment. A small sample size with large standard deviations between approaches makes it fundamentally hard to draw the strong conclusions you are making.

Improvements:

4. The source for the FoV overlap definition (equation 6) is missing, it should be cited; alternatively the formula should be derived or motivated (briefly). Prior work uses different way of computing FoV overlap (e.g., Yu et al, 2025, use intersections between four rays from two camera origins); some brief discussion on the definition used is arguably warranted.

5. The choice of hyperparameters (e.g., compression ratios, temporal penalty weights) is not explained (e.g., where there preliminary experiments or prior work that guided the selection?), please consider adding some discussion, even if brief.

6. Cite relevant prior art on dynamic per-frame compression allocation ([1, 2]) and spatial memory ([3])
	[1] ElasticTok, Yan et al, 2025
	[2] CAT, Shen et al, 2025 (image only)
	[3] Video World Models with Long-term Spatial Memory, Wu et al, 2025

7. The second paragraph starts with "Recent work has begun to address these challenges", but the manuscript has not listed any challenges yet.

---

> ### Author Response · Authors · 2026-07-22
>
> Thank you for insightful and helpful reviews.
>
> > The reported results completely omit statistics beyond the aggregates (e.g., standard deviation across the eval set). E.g., LoopNav reports everything as mean +- stdev. I strongly encourage authors to not drop stdev, especially given that empirical results are a big part of the contribution.
>
> > Explicitly acknowledge the limitations of the experiment. A small sample size with large standard deviations between approaches makes it fundamentally hard to draw the strong conclusions you are making.
>
> Thank you for pointing this out. In the revised manuscript, following the original LoopNav paper, we now report the mean and standard deviation across evaluation trajectories for all PSNR, LPIPS, SSIM, and DreamSim results in all tables.
>
> Note that LoopNav is a challenging benchmark with diverse trajectories; thus, trajectory-level variation is substantial. However, as shown in **Table 1**, the standard deviations of WorldPack are broadly comparable to those of the other baselines.
>
> FVD, in contrast, is defined as a distributional distance between sets of videos. The public LoopNav code appears to compute FVD separately for each trajectory and then reports the variance across those values (https://github.com/Kevin-lkw/LoopNav/blob/main/baseline/metrics/calcmetric.py), but this procedure captures a different quantity from the standard set-level definition of FVD [Unterthiner+ 2019]. We therefore follow the original FVD formulation and report a single distributional distance computed over the corresponding set of evaluation videos for each task–range condition.
>
> > The LoopNav benchmark itself has minimal external adoption (no peer reviewed work that I know uses it), which makes it hard to assess how informative the empirical results are.
>
> We use LoopNav because it is specifically designed to evaluate long-term spatial memory. Its loop-structured trajectories require an agent to return to previously observed locations, thereby directly testing whether a model can retain and reconstruct past observations. We have clarified this motivation in **Section 5**.
>
> > Please explicitly state the number of evaluation trajectories in all experiments. If you use only 18 in the main experiment, please use more of the test set (there appears to be 960 test trajectories available).
>
> The main evaluation does not use only 18 trajectories. It uses 144 trajectories in total across all task types (ABA, ABCA) and navigation ranges (5, 15, 30, 50), with 18 trajectories per setting. It is consistent with the evaluation set configuration reported in the original LoopNav paper, thereby providing a basis for fair comparison. We have explicitly added these details to **Appendix C.1**.
>
> > Other benchmarks have more than an order of magnitude larger evaluation sets (e.g., Mineworld 1000 test clips, WorldMem 300 test videos). Note that in video world modelling, the fact that frames within a video are highly correlated makes it necessary to increase trajectory sample size, rather than only increasing the frame-level sample size (i.e., more videos is valuable and in that sense orthogonal to longer videos).
>
> In **Appendix C.1**, we now compare LoopNav, MineWorld, and WorldMem in terms of the number of evaluation videos, the total number of evaluated frames, and the maximum rollout length per trajectory.
>
> We agree that the number of independent trajectories and rollout length capture complementary dimensions of evaluation. Although LoopNav uses fewer evaluation videos than MineWorld and WorldMem, its 144 trajectories are within an order of magnitude of those used by either benchmark. Moreover, LoopNav features substantially longer rollouts and evaluates a total number of frames comparable to WorldMem. We therefore consider the evaluation comparable in overall scale, while emphasizing a distinct dimension: long-horizon spatial-memory evaluation rather than short-clip generation.
>
> | Benchmark | # Evaluation videos | Max. video length | # Total frames |
> | --- | --- | --- | --- |
> | MineWorld (Guo et al., 2025) | 1,000 | 16 | 16,000 |
> | WorldMem (Xiao et al., 2025) | 300 | 100 | 30,000 |
> | LoopNav (Lian et al., 2025)  | 144 | 630 | 30,738 |
>
> [Unterthiner+ 2019] Towards Accurate Generative Models of Video: A New Metric & Challenges.

---

> > ### Author Response · Authors · 2026-07-22
> >
> > > If you use only the 18 eval trajectories mentioned above in the main experiment, this exacerbates the lack of standard deviation reporting. Furthermore, FVD might be a poor choice at this sample size.
> >
> > As noted above, the full main evaluation uses 144 trajectories, and we now report the mean and standard deviation across evaluation trajectories for all PSNR, LPIPS, SSIM, and DreamSim results.
> >
> > However, the FVD values in the main text were computed separately for each task type and navigation range, so each individual FVD value was based on 18 trajectories.
> >
> > To provide an FVD evaluation across a larger set of videos, we also report FVD computed jointly across all 144 trajectories in **Appendix G**. Under this evaluation, WorldPack again outperforms the Baseline, Nearest Frame Packing, and Memory Retrieval, indicating that the improvement persists when FVD is computed over a larger video set.
> >
> > | Method | FVD ↓ |
> > | --- | --- |
> > | Baseline | 872 |
> > | Nearest Frame Packing | 846 |
> > | Memory Retrieval | 676 |
> > | **WorldPack (ours)**  | **608** |
> >
> >
> > > The use of RECON does not solve this problem, because (1) the sample size is not reported, and (2) that experiment is itself more limited as it only compares with a single baseline.
> >
> > For the RECON experiment, we have added explicit details on the number of trajectories, data split, context configuration, and generation protocol in **Appendix C.2**.
> >
> > The RECON experiment serves as complementary validation on real-world data, while the main LoopNav evaluation provides broader comparisons against multiple existing methods and component-level baselines. We have clarified this experimental role and explicitly identified the RECON baseline as NWM.
> >
> > > The source for the FoV overlap definition (equation 6) is missing, it should be cited; alternatively the formula should be derived or motivated (briefly). Prior work uses different way of computing FoV overlap (e.g., Yu et al, 2025, use intersections between four rays from two camera origins); some brief discussion on the definition used is arguably warranted.
> >
> > The FoV overlap definition and its Monte Carlo computation are based on the implementation of WorldMem by Xiao et al. To clarify the attribution, we have added the corresponding citation to the Geometric Selection subsection in **Section 4.2**.
> >
> > > The choice of hyperparameters (e.g., compression ratios, temporal penalty weights) is not explained (e.g., where there preliminary experiments or prior work that guided the selection?), please consider adding some discussion, even if brief.
> >
> > We have added the rationale for the hyperparameter choices to the Geometric Selection and Implementation Details subsections in **Section 4.2**. Specifically, the 10,000 Monte Carlo samples used for FoV overlap estimation and the weighting coefficients w_o=1.0 and w_t=0.2 follow WorldMem. The three compression rates are based on the rate-specific projection design of FramePack. We have also added further details on the learning rate, optimization settings, model initialization, and dataset configuration to **Appendices C** and **D**.
> >
> > > Cite relevant prior art on dynamic per-frame compression allocation ([1, 2]) and spatial memory ([3]) [1] ElasticTok, Yan et al, 2025 [2] CAT, Shen et al, 2025 (image only) [3] Video World Models with Long-term Spatial Memory, Wu et al, 2025
> >
> > Thank you for the suggestion. We have added citations to ElasticTok, CAT, and Video World Models with Long-term Spatial Memory in **Section 1**, and clarified how our work relates to prior studies on dynamic frame- or region-level compression allocation and long-term spatial memory.
> >
> > > The second paragraph starts with "Recent work has begun to address these challenges", but the manuscript has not listed any challenges yet.
> >
> > We agree that the original wording was unclear because "these challenges" was referenced before the challenges had been explicitly introduced. We revised the relevant part of **Section 1**.

---

### Review · Reviewer_GG3Y · 2026-06-21

**Summary Of Contributions:**

This work addresses temporally and spatially consistent long-horizon video generation for world models. Prior approaches broadly fall in two buckets: (i) temporal proximity: compress past frames at varying rates (depending on temporal proximity), (ii) spatial memory retrieval: select past frame based on field-of-view overlap. (i) struggles in cases where a temporally distant frame has 3D overlap with next frame to generate while (ii) operate within a fixed context window.

WorldPack treats frame selection and compression together: frames that have strong 3D overlap with current frame are retained at higher resolution whereas less relevant frames are strongly compressed but not entirely discarded. This expands the effective context from 4 to 22 frames (5.5x rise) while only increasing inference time by 16% and memory by 12%.

WorldPack is evaluated on LoopNav benchmark based on Minecraft videos and a real-world benchmark for navigation. It achieves strong results beating baselines such as NWM, DIAMOND.

**Strengths**

1. The method is simple and well-explained: it extends FramePack (which choses context frames by temporal recency) to use a geometric-measure of how strongly a past frame overlaps with current frame in 3D.
2. The experiments show effectiveness of WorldPack: On both benchmarks, the results are convincing (except for variations in SSIM which is not an ideal metric for perceptual fidelity). The ablation study is also well-designed.
3. The paper is well-written and was easy to understand.

**Weaknesses**

1. The measure of overlap between past and current frame needs camera pose annotations for every frame. For a general real-world setting, this seems like a strong assumption. This hasn't been discussed sufficiently.
2. Limited training details: I think the paper does not provide sufficient details on the dataset, hyperparameters, etc. These should at least be included in the Appendix.
3. At each step, the method computes FoVOverlap between all the past frames and the current frame. This seems an expensive operation. How is this managed? Is there a way to reuse/cache intermediate results to accelerate inference?
4. [Minor] I think the image used to highlight "Low Score" in Fig 1 (Trajectory Packing) is the same as "High Score" which seems like an error.

**Audience:**

Yes

**Audience Explanation:**

World Models are quite important and have implications for simulation, planning, robotics, etc. This paper leads an effort to make more effective use of the past context frames leading to better long-horizon generation. Some of the learnings can also apply to general video generation as well. Thus, it will be of interest to the video generation community and world modelling community.

**Claims And Evidence:**

Yes

**Claims Explanation:**

The central claim is that using combination of geometry-based frame selection in trajectory packing leads to better world models. The approach makes intuitive sense and is also shown experimentally to work better.

- WorldPack is evaluated on one synthetic (LoopNav based on Minecraft) and one real-world benchmark (navigation). The experiments clearly show its effectiveness.
- The uptick in inference compute and memory are quite reasonable given the benefit of using 5.5x more context frames.

**Requested Changes:**

1. The authors must clarify that camera poses are assumed to be available and discuss how feasible it is in a real-world setting, i.e., the gains do not come for free since the FoVOverlap needs camera poses.
2. More details on the training pipeline should be given.
3. While the authors state 16% uptick in inference compute, a more thorough explanation of FoVOverlap compute cost should be included. How can this method scale to increase context further?

---

> ### Author Response · Authors · 2026-07-22
>
> Thank you for your constructive and insightful comments.
>
> > The measure of overlap between past and current frame needs camera pose annotations for every frame. For a general real-world setting, this seems like a strong assumption. This hasn't been discussed sufficiently.
>
> > The authors must clarify that camera poses are assumed to be available and discuss how feasible it is in a real-world setting, i.e., the gains do not come for free since the FoVOverlap needs camera poses.
>
> In **Section 7**, we added an explicit discussion of the limitation that WorldPack assumes access to the camera poses of both the current and historical frames. We also explain that, in real-world settings, these poses can be estimated using established methods such as simultaneous localization and mapping (SLAM) or visual-inertial odometry. However, pose estimation errors may affect frame selection based on field-of-view overlap. We therefore clarify that the performance gains achieved by our method come with the additional cost of obtaining or estimating camera poses. We further identify evaluating robustness to pose estimation errors and exploring feature-based or implicit pose estimation as important directions for future work.
>
> > Limited training details: I think the paper does not provide sufficient details on the dataset, hyperparameters, etc. These should at least be included in the Appendix.
>
> > More details on the training pipeline should be given.
>
> To improve reproducibility, we additionally report the hyperparameters used for FoV overlap computation, including the number of Monte Carlo samples and the weighting coefficients, in **Section 4.2**. We provide the dataset and evaluation configurations in **Appendix C**, and add a detailed description of the training pipeline, including the training procedure, optimization settings, and model initialization, in **Appendix D**.
>
> > At each step, the method computes FoVOverlap between all the past frames and the current frame. This seems an expensive operation. How is this managed? Is there a way to reuse/cache intermediate results to accelerate inference?
>
> > While the authors state 16% uptick in inference compute, a more thorough explanation of FoVOverlap compute cost should be included. How can this method scale to increase context further?
>
> Regarding computational cost, the previously reported 16% increase reflects the diffusion model's inference time, excluding the computation of FoV overlap. This value reflects the additional cost of the diffusion-model component when trajectory packing increases the number of visible frames from 4 to 22. We have clarified this point in the revised manuscript. The runtime scales approximately linearly with the number of candidates, indicating predictable computational growth as the accessible context is expanded.
>
> We further report the runtime of FoV-based geometric selection for different numbers of memory candidates in **Table 6**: 0.05, 0.06, 0.10, and 0.26 seconds for 50, 100, 400, and 1,600 candidates, respectively. These results show that the additional cost remains moderate for several hundred candidates, while also making clear that it increases with the number of candidates as the accessible context expands.
>
> > I think the image used to highlight "Low Score" in Fig 1 (Trajectory Packing) is the same as "High Score" which seems like an error.
>
> Thank you for pointing this out. We fixed the figure.

---

### Review · Reviewer_8xx5 · 2026-07-10

**Summary Of Contributions:**

The paper proposes WorldPack, a video-generative world model targeting long-horizon spatial consistency. The core idea is to combine two existing lines of work: FramePack-style hierarchical frame compression and WorldMem-style geometric frame scoring. WorldPack allocates compression rates by spatial relevance, measured as the overlap in the viewing frustum with the current pose. The contribution is not the spatial scoring itself but using it to modulate the FramePack compression rates. Experiments on the LoopNav Minecraft benchmark and on RECON show improvements over the baselines.

**Audience:**

Yes

**Audience Explanation:**

Long-horizon consistency in interactive video world models is an active area (WorldMem, Context-as-Memory, RELIC, HY-World 1.5's "reconstituted context memory" all appeared in 2025). The finding that spatially-guided graded compression of a large history beats both packing based on temporal proximity and fixed-window spatial retrieval under an equal token budget (Table 3) is a clean, actionable insight.

**Broader Impact Concerns:**

None beyond what is already covered. The included Broader Impact Statement is adequate for this scope.

**Claims And Evidence:**

No

**Claims Explanation:**

1. The context lengths of the baselines in Table 1 are misreported. All baseline numbers in Table 1 are copied verbatim from Lian et al. (2025), but the submission changes Mineworld's context length from 32 to 15 and DIAMOND's from 32 to 4. The submission also drops Lian et al.'s reported standard deviations.
2. Provenance of the values in Table 2 is unspecified. Lian et al. don’t report PSNR or DreamSim for NWM.
3. The values reported in Table 3 are inconsistent with those reported for ABA-5 in Tables 1 and 2. If these come from different checkpoints or evaluation subsets, this must be stated
4. The math in the implementation details paragraph is inconsistent. “$\lambda = 2$ with $d_i ∈ \{0,1,2\}$" yields ratios 1, 2, 4 under Eq. (4), not the stated $2^0, 2^2, 2^4$. Also, $L_f$ is not reported.
5. Equations 6 and 7 reproduce WorldMem's retrieval score, yet no attribution is made at the point of definition.
6. The paper frames FramePack as doing compression only based on temporal proximity. FramePack already contemplates non-temporal (feature-similarity/hybrid) importance measures, which should be acknowledged. WorldPack is an instantiation of FramePack that scores compression on spatial overlap for pose-conditioned world models, which is a reasonable delta.

**Requested Changes:**

Critical (required for me to recommend acceptance)
1. Correct the baseline context lengths in Table 1 to match Lian et al. and restore the standard deviations.
2. Clarify the provenance of the values in Table 2.
3. Resolve the WorldPack ABA-5 inconsistency between Tables 1/2 and Table 3
4. Fix the values in the implementation details paragraph so that they are consistent with Eq. (4), and state $L_f$ explicitly.
5. Attribute the FoV-overlap score of Eqs. (6) - (7) to WorldMem at the point of definition.
6. Acknowledge that FramePack also proposes non-temporal importance metrics.


Strengthening (not required)
1. Report a comparison on RECON against NWM, which is the source of the CDiT backbone.
2. An analysis of failure cases under occlusion (acknowledged limitation) and a sensitivity study over $(S, \lambda, d_i)$ schedules would considerably deepen the paper's insight beyond the single configuration reported.
3. Remove the duplicated citation for Imagen.

---

> ### Author Response · Authors · 2026-07-22
>
> Thank you for the detailed and constructive review.
>
> > Correct the baseline context lengths in Table 1 to match Lian et al. and restore the standard deviations.
>
> The baseline context lengths in **Table 1** were originally taken from version 1 of the LoopNav arXiv paper [Lian+ 2025]. Because the description of the context-length settings was updated in the latest version, v3, we revised the values in **Table 1** accordingly.
>
> Following the evaluation protocol of the original paper, we also recomputed PSNR, LPIPS, SSIM, and DreamSim by first averaging each metric within each trajectory and then reporting the mean and standard deviation across trajectories.
>
> > Clarify the provenance of the values in Table 2.
>
> The values in **Table 2** were obtained from our trained NWM baseline and WorldPack models. We now state this explicitly in **Table 2** and provide the corresponding training recipe and implementation details in **Appendix D**.
>
> > Resolve the WorldPack ABA-5 inconsistency between Tables 1/2 and Table 3.
>
> Thank you for catching this. Some entries mistakenly referred to different evaluation outputs. We corrected the affected values and verified that **Tables 1–3** now consistently refer to the same evaluation results.
>
> > Fix the values in the implementation details paragraph so that they are consistent with Eq. (4), and state  explicitly.
>
> We revised the Implementation Details paragraph in **Section 4.2** to $d_i\in\{0,2,4\}$, ensuring that the reported compression ratios and context lengths are consistent with Eq. (4). We also explicitly state the relationship between each compression ratio and the number of frames assigned to it.
>
> > Attribute the FoV-overlap score of Eqs. (6) - (7) to WorldMem at the point of definition.
>
> We now explicitly attribute the FoV-overlap score in Eqs. (6)–(7) to WorldMem at its point of definition in the Geometric Selection subsection of **Section 4.2**.
>
> > Acknowledge that FramePack also proposes non-temporal importance metrics.
>
> FramePack proposes not only temporal-proximity-based importance, but also frame-similarity-based importance and a hybrid of the two. We revised the **Abstract**, **Introduction**, **Related Work**, and **Method** sections to represent these contributions accurately.
>
> > Report a comparison on RECON against NWM, which is the source of the CDiT backbone.
>
> We agree that the identity of the RECON baseline was unclear. The baseline reported in **Table 4** is NWM, from which WorldPack adopts the CDiT backbone. We have now clarified that the baseline reported in the RECON table is NWM, from which the CDiT backbone is adopted. We also provide the evaluation configuration in **Appendix C** and the corresponding training schedule and optimization settings in **Appendix D**.
>
> > An analysis of failure cases under occlusion (acknowledged limitation) and a sensitivity study over  schedules would considerably deepen the paper's insight beyond the single configuration reported.
>
> The sensitivity of WorldPack to the packing schedule is discussed in **Appendix E**. In particular, the analysis shows that overly aggressive compression can degrade generation quality and suggests that heavily compressed frames may retain insufficient visual information. This discussion clarifies that increasing the compression ratio does not necessarily improve performance and that an appropriate balance between historical coverage and retained fidelity is required.
>
> We also expanded **Section 7** to discuss failure modes under occlusion. Because the FoV-overlap score is derived from camera poses and viewing frustums, it may overestimate relevance when geometry overlaps, but the corresponding scene content is occluded. We identify this as a limitation and a direction for developing visibility-aware importance measures.
>
> > Remove the duplicated citation for Imagen.
>
> We removed the duplicated citation to Imagen.
>
> [Lian+ 2025] LoopNav: Benchmarking Spatial Consistency in World Models